# Evaluation of Alternative Doxycycline Antibiotic Regimes in an Inhalational Murine Model of Q Fever

**DOI:** 10.3390/antibiotics12050914

**Published:** 2023-05-16

**Authors:** Kate A. Clay, M. Gill Hartley, Adam O. Whelan, Mark S. Bailey, Isobel H. Norville

**Affiliations:** 1Academic Department, Royal Centre for Defence Medicine (Academia and Research), Birmingham B15 2GW, UK; 2CBR Division, Defence Science and Technology Laboratory (Dstl), Porton Down, Salisbury SP4 0JQ, UK; 3Department of Biosciences, University of Exeter, Stocker Road, Exeter EX4 4QD, UK

**Keywords:** *Coxiella burnetii*, antibiotic, mouse model, doxycycline, Q fever

## Abstract

The timing of the initiation of antibiotic treatment has been shown to impact the clinical outcome of many bacterial infections, including Q fever. Delayed, suboptimal or incorrect antibiotic treatment has been shown to result in poor prognosis, resulting in the progression of acute disease to long-term chronic sequalae. Therefore, there is a requirement to identify an optimal, effective therapeutic regimen to treat acute Q fever. In the study, the efficacies of different doxycycline monohydrate regimens (pre-exposure prophylaxis, post-exposure prophylaxis or treatment at symptom onset or resolution) were evaluated in an inhalational murine model of Q fever. Different treatment lengths (7 or 14 days) were also evaluated. Clinical signs and weight loss were monitored during infection and mice were euthanized at different time points to characterize bacterial colonization in the lungs and the dissemination of bacteria to other tissues including the spleen, brain, testes, bone marrow and adipose. Post-exposure prophylaxis or doxycycline treatment starting at symptoms onset reduced clinical signs, and also delayed the systemic clearance of viable bacteria from key tissues. Effective clearance was dependent on the development of an adaptive immune response, but also driven by sufficient bacterial activity to maintain an active immune response. Pre-exposure prophylaxis or post-exposure treatment at the resolution of clinical signs did not improve outcomes. These are the first studies to experimentally evaluate different doxycycline treatment regimens for Q fever and illustrate the need to explore the efficacy of other novel antibiotics.

## 1. Introduction

*Coxiella burnetii*, the causative organism of Q fever, is an intracellular Gram-negative bacterium that is endemic worldwide and survives for long periods in the environment. It has a very low infectious dose, with a single bacterium calculated as being able to cause disease. *C. burnetii* is shed in huge quantities in ruminant birth products, for example, infected goat placentas contain 10^9^ genome equivalents (GE)/gram of tissue [1]. The majority of human infections are by the inhalation of contaminated particles, and therefore Q fever presents a significant hazard to military personnel deployed in dry, dusty areas such as Afghanistan, where farmed ruminants are ubiquitous [2]. Q fever is difficult to diagnose (requiring multiple blood samples and specialized laboratory facilities), and acute infections can significantly affect a soldier’s performance (in the short term) but also have the potential to lead to long-term chronic complications. Therefore, the efficacious treatment of both acute and chronic disease is a high priority for military operations, and as personnel are often deployed to high-risk areas, the possibility of chemoprophylaxis should be considered [2]. Delord et al. suggested that doxycycline as chemoprophylaxis for malaria prevention will prevent most rickettsial diseases in travelers, although this is not evidence-based [3]. To date, there have been no published data to support the efficacy of antibiotic treatment in Q fever prevention.

First-line treatment (within the UK) for acute Q fever is doxycycline hyclate, typically given for 14 days and only started once an individual has symptoms [4]. Tetracycline administration within the first three days of illness can reduce the duration of the fever by up to 50% [5]. Doxycycline hyclate is associated with side effects such as gastrointestinal irritation and skin sensitivity, which often lead to incomplete treatment courses. Doxycycline monohydrate has been shown to be better tolerated [6] but is more expensive in the UK. There is a suggestion that the inadequate treatment in the acute phase can increase the chance of progression to chronic Q fever [7]. A retrospective analysis of the UK military Q fever cohort returning from Afghanistan between 2009 and 2014 suggests a delay in the initiation of doxycycline for greater than 5 days after symptom onset is associated with an increased likelihood of developing Q fever fatigue syndrome (unpublished data). Delayed post-exposure therapy has been advocated based on a single human paper showing doxycycline treatment should be delayed until the onset of symptoms to prevent relapse following treatment completion and to allow full immunity to develop [8]. However, there have been no large human or animal trials investigating the optimum timing for the initiation of antibiotic therapy after exposure to *C. burnetii*.

Approximately 2% of infections with *C. burnetii* lead to chronic infection and 20% of symptomatic infections lead to Q fever fatigue syndrome [7]. To reduce the likelihood of developing chronic infection, the effective treatment of Q fever could be judged not just by the relief of symptoms, but also on systemic clearance and allowing the development of protective immunity. A study following the cardiac surgery of individuals known to have had Q fever found increased antibody levels in 17% of individuals but no illness, suggesting a resurgence of the infection controlled by an effective immune response [9]. Acute Q fever has been modelled in a large range of different animals [10], none of which fully represent human disease. The most widely used models are various strains of mice where disease is often mild and self-limiting, e.g., A/J male mouse [11], and the BALB/C mouse [12], or the Guinea pig, where the disease is frequently fatal [13]. Improved models are perhaps best represented in non-human primates, e.g., rhesus macaques [14] or marmosets [15], which have closer immunological and genetic similarities to humans [16]. Previously, using the A/J mouse model of Q fever, we have demonstrated that 7 days of treatment with doxycycline hyclate initiated at 24 h post-exposure reduced clinical signs compared to a PBS control [17]. However, it did not decrease bacterial burden in either the lungs or spleen at the end of the study (day 14), as assessed by PCR. Despite the apparent initial resolution of disease, data from our model also showed body weight loss towards the end of our study, suggesting a resurgence of the infection following the cessation of treatment due to the bacterial static nature of doxycycline. Doxycycline hyclate was also poorly tolerated by the mice, causing drug-associated weight loss, complicating the analysis of weight data and occasionally resulting in mice reaching the humane end point.

In this study, we used the A/J mouse model to evaluate the alternative, better-tolerated formulation of doxycycline: doxycycline monohydrate. Experimentally, both the minimum inhibitory concentration (MIC) and the in vitro efficacy in a cell infection assay were comparable between the two formulations (MIC: <0.04 µg/mL, in vitro: 0.2 µg/mL) [18]. We also extended the treatment time to 14 days to better mimic human treatment regimens and evaluated the impact of alternative treatment timings (pre-exposure prophylaxis, post-exposure prophylaxis, treatment at onset of clinical signs and after resolution of clinical signs). These timings were chosen to represent real-life scenarios, for example, military personnel are sometimes prescribed doxycycline as anti-malarial prophylaxis and therefore might already be taking doxycycline when exposed (pre-exposure prophylaxis). The initiation of treatment at day 2 post-exposure represented preventative treatment when a probable source has been located, such as a lab-acquired infection or outbreak at a local goat farm. Day 5 post-exposure treatment represented the onset of clinical symptoms, the point at which someone might seek treatment, and day 10 was included to determine if clearance could be improved by treating after the resolution of primary clinical symptoms. These treatments were assessed for their ability to achieve the following: prevent or ameliorate disease (weight loss and clinical signs); moderate the development of adaptive immunity (cell-mediated response and antibody production); and control the ongoing innate immune response (circulating IFNγ levels). Viable counts made at key time points were used to monitor bacterial clearance and recovery versus resurgence or reinfection in both the lung (as the primary site of infection) and the spleen (as an indicator of systemic clearance). 

Furthermore, the dissemination of viable bacteria into other organs of interest was also assessed, and these included the brain, testes, bone marrow and adipose tissues. The brain was included because, in human disease, severe headaches are a common feature of acute Q fever, possibly due to febrile illness and dehydration, but neurological complications such as meningitis and meningoencephalitits have also been documented [19]. The testes were assessed to gain further information on the sexual transmissibility of the bacteria [20]. Confocal imagery was used to determine how sperm cells, which have minimal cytoplasm, could harbor the *C. burnetii* bacterium. *C. burnetii* DNA is commonly found in bone marrow biopsies at all stages of the disease [21,22], and both bone marrow and adipose tissue have been postulated as potential sanctuary sites for recrudescence of the disease [23,24].

## 2. Results

### 2.1. Dissemination of Bacteria Following Aerosol Challenge in the A/J Mouse Model of Q Fever

As part of study one, the A/J mouse model was further developed and characterized in order to more fully evaluate antibiotic treatment and clearance. Mice were challenged via the aerosol route with a mean presented dose of 1 × 10^7^ GE *C. burnetii* (calculated to be a mean retained dose of 6.7 × 10^4^ GE) and treated with water for 14 days post-challenge. Clinical signs were monitored, and bacterial dissemination characterized in seven tissues at six time points post-challenge. Mice lost weight peaking at day 8 and recovering by day 11 (Figure 1). There was also evidence of the failure to gain weight due to the handling associated with the water treatment which occurred before the typical *C. burnetii*-associated weight loss onset at day 5 PC. All mice showed some clinical signs of infection (ruffled fur and occasional hunched appearance) from day 5 to 12. Groups of five mice were euthanized at day 5, 8, 12, 15, 19 and 35 post-challenge (PC), and organs aseptically removed. Lung weights peaked at day 8 but were normal by day 15. Spleen weights peaked between day 8 and 12 and were still enlarged at day 35 (mean 0.394% of bodyweight day 35 PC vs. 0.291% of bodyweight uninfected control *p* < 0.05). The lungs and spleens had the highest viable counts on day 5 to 8 PC. The counts in the spleens declined rapidly with the majority of animals having cleared viable bacteria from their spleens by day 19 PC. Clearance from the lungs took much longer, with viable bacteria isolated from two of the five animals on day 35 PC.

Viable bacteria were isolated from all the tissues tested (Figure 2), with the highest proportion of mice infected on day 8 PC. All mice had bacteremia at the peak of infection, with the majority having PCR-positive blood until at least day 12. Viable bacteria were isolated from the majority of brain tissue samples, and for an extensive period, from the earliest time point tested (day 5 PC) until at least day 12 or 15 PC. Additionally, viable bacteria were found within the testis homogenates. A visual (microscopic) examination detected excessive extracellular material on the sperm head (Figure 3B) compared to the smooth finish of the healthy uninfected sperm (Figure 3A). The sperm themselves appeared uninfected, with *C. burnetii* bacteria visible in the accompanying white cells observed in two mice on day 8 PC that were also culture-positive (Figure 3C,D). However, the inflammatory response of extracellular debris was observed in the majority of samples taken during the peak of infection days 5–8.

### 2.2. Evaluation of Alternative Doxycycline Treatment Lengths (7 vs. 14 Days) Initiated Pre- or Post-Challenge in the A/J Mouse Model of Q Fever (Study 1)

The efficacy of doxycycline therapy initiated at either symptom onset (day 5 PC) or as a pre-exposure prophylaxis (one day prior to challenge) was compared to a water-treated control group (WC). Additionally, the effect of different treatment lengths on disease outcome was assessed by treating infected mice for either 7 or 14 days. All treatment regimens effectively prevented or ameliorated infection-associated weight loss (Figure 4). Mice given pre-exposure prophylaxis suffered a small (2%) weight loss associated with treatment before challenge, but they experienced no further weight loss (significantly different from the WC group from day 6 to day 10). However, they did display some fur ruffling from days 7 to 8 PC. The mice starting treatment on day 5 were already showing some clinical signs, such as fur ruffling and weight loss. The weight loss stopped on day 6 PC (significantly different from WC on days 7 to 9), and fur ruffling stopped by day 10. There was no difference in weight gain or clinical signs between 7 and 14 days treatment, but those treated prior to exposure gained weight quicker from days 18 to 28 PC (significant for the 7 day pre-exposure treatment group only).

Systemic spread was determined by the enumeration of bacterial colonization in tissues from mice euthanized at key time points, either 24 h after the completion of respective 7-day or 14-day therapy regimes, or at the end of the study (Figure 5A,B). Viable bacteria were found in the spleens and lungs but not in other tissues (brain, bone marrow, testis, adipose tissue nor blood). Splenomegaly was only significant in the pre-exposure, 7 days of treatment group, and only on day 35 PC (compared to naïve *p* < 0.05), despite the lack of cultivable bacteria. On day 15, following 14 days of treatment starting pre-exposure, it was not possible to culture bacteria from the spleens and only 3/5 mice had infected lungs. However, by day 35, there was some evidence of resurgence of the disease, 3/8 mice had infected spleens and 7/8 had infected lungs. Despite this, there were no accompanying clinical signs. It is possible that bacteria were also able to spread systemically during treatment in the post-treatment regime (1/5 mice on day 12 had infected spleens compared to 1/4 by day 19). In the same period, the water-treated control group had achieved a marked improvement with the number of mice with infected spleens declining from 5/5 to 1/5, suggesting that doxycycline treatment might have inhibited systemic clearance. 

The levels of circulating IFNγ (in the blood) were determined at each of the time points (Figure 5C). Pre-exposure prophylaxis prevented any bacterial growth or activity which resulted in no elevation of IFNγ above the naïve control levels (day 8 pre-7D, Figure 5C). Treatment starting on day 5 reduced the level of circulating IFNγ detected on day 12 to 1/10 of that detected for the WC group, which may have been sufficient to reduce the innate immune response.

On day 35, anti-phase I coxiella IgG titers were measured (Figure 5D). All infected groups raised significant levels of anti-*Coxiella* antibodies compared to naïve (*p* < 0.001); however, the levels achieved by the group receiving 16 days of treatment including pre-treatment had significantly lower antibody levels than the water-treated group (*p* < 0.05).

### 2.3. Impact of Initation of Doxycycline Treatment at Different Times Pre- and Post-Challenge on Clinical Signs in the A/J Mouse Model of Q Fever (Study 2)

Study one demonstrated that 14 days of treatment initiated at symptom onset reduced bacterial load and dissemination which may result in the reduced likelihood of long-term colonization and chronic disease. In order to further refine the optimum post-exposure treatment window, additional time points (day 2, 5, 10 PC) for the initiation of therapy were compared in study 2. Mice were euthanized 14 days after the completion of treatment (day 28 to 35 PC).

To further determine the effect of oral dosing on weight loss in this model, a group of unchallenged water-treated mice was included. This control treatment prevented the weight gain that normally occurs in uninfected mice, presumably a detrimental effect of handling. Once the treatment stopped, on day 16 (Figure 6A black line) the mice started to gain weight in the expected manner. Infection combined with water treatment resulted in *coxiella*-associated weight loss from day 6 to day 12 PC (Figure 6A red line, significant on days 8, 9 and 10 PC), and other characteristic signs of disease, such as ruffled fur, from day 3 to 13 PC (Figure 6C). This group also did not properly start to gain weight until the treatment and handling had stopped. Following the cessation of treatment, the two groups gained weight in a comparable manner. 

Doxycycline treatment starting at either day 2 (blue line, Figure 6A) or day 5 PC (green line, Figure 6A) was effective at controlling weight loss from infection, with neither group suffering more weight loss than that experienced by the unchallenged water-treated control group (Figure 6A). However, the majority of mice showed some signs of illness: ruffled fur or unkempt appearance from day 4/5 until day 17 when treatment started on day 2, and marginally worse (from day 7 to 30) when treatment started on day 5, although for either group, these signs were reduced compared to the infected control mice (Figure 6C). Once treatment stopped, the animals gained weight in the normal manner.

Starting treatment late in the infection, on day 10 PC, further illustrated the detrimental effect that the handling had on the mice, although the percentage of weight loss attributable to disease (that lost between day 5 and 8) remained unaffected (Figure 6B). The clinical signs for this group, although marginally better than for the infected control group, continued for the entire duration of their treatment (day 26) (Figure 6C).

The prophylaxis treatment starting one day prior to the challenge resulted in greater initial weight loss than that observed in either of the water treatment groups (significant on the day of challenge and day 6 PC *p* < 0.05), prior to the onset of the effect from infection (Figure 6B). However, doxycycline appeared to prevent weight loss from infection, with the mice slowly gaining weight from day 5, although they had ruffled fur from day 3 to day 16. Once treatment stopped, this group recovered at a comparable rate to the control groups.

### 2.4. Impact of Initation of Doxycycline Treatment at Different Times Pre- and Post-Challenge on Bacterial Clearance in a Mouse Model of Q Fever

At the end of the study, 14 days after the end of treatment, the animals were euthanized. There was some evidence of splenomegaly in the pre-treated (mean 0.427% of bodyweight *p* < 0.05) mice and those whose treatment started on day 10 (0.431% of bodyweight *p* < 0.01) compared to the naïve controls (0.291% of bodyweight), but not compared to the water-treated infected controls (0.357% of bodyweight). Treatments starting day 2 or 5 did not result in measurable splenomegaly 28 days later. Spleen and lung samples were assessed for viable bacteria and indicated that treatment with doxycycline may have had a detrimental effect on bacterial clearance, in particular in the lung (Figure 7A). Starting treatment before the challenge did not prevent systemic spread, with this group fairing worse than those starting treatment on day 2. The initiation of treatment on day 5, at the onset of clinical signs, resulted in poorer clearance than the initiation of treatment on day 2, perhaps as a result of the much higher counts by the unchecked bacterial replication at the time of starting treatment. The lungs from the treatment start on day 5 were significantly more colonized than those from the treatment start on day 2, despite having 3 days longer to clear (2.56 × 10^3^ cfu/g vs. 6.78 × 10^2^ cfu/g *p* = 0.038). The water-treated group appeared to have better clearance than those whose treatment started on day 10, with 6/6 lungs still infected and a mean count of 3.19 × 10^2^ cfu/g. 

Compared to study 1, treatment initiation on day 5 was also detrimental to bacterial clearance from the spleen. In total, 4/6 spleens were infected with a mean of 2.06 × 10^3^ cfu/g on day 33, whereas 4/5 spleens from the WC group (from study 1) were clear (below LOD) by day 19 and all by day 35.

The spleens were also used as a source of T cells, to assess the presence of any specific cell-mediated immunity. Too few cell populations survived well enough to assess the activity in the prophylaxis group, but in all other treatment groups, all animals produced IFNγ following exposure to the heat-killed phase I coxiella. The majority of animals regardless of treatment start produced more IFNγ to heat-killed coxiella than to the control stimulant ConA (Figure 7B), suggesting a specific adaptive immune response, compared to the naïve animals which were more responsive to ConA than inactivated coxiella.

## 3. Discussion

The time point at which antibiotic therapy is initiated is critical to ensuring the effective amelioration of clinical symptoms, clearance of the bacterial infection and prevention of latent infection or relapse. If the antibiotic is given too late, then the bacteria may have disseminated, colonized a range of tissues and caused bacteremia, sepsis or even death. In the case of Q fever, a delay in treatment has also been shown to result in a higher likelihood of serious chronic disease, such as Q fever chronic fatigue syndrome or endocarditis [7,22,25]. If the antibiotic is administered inappropriately, either too early or at a high concentration, then side effects may be observed, including gastrointestinal issues, skin sensitivity or immune modulation [6]. The studies reported here experimentally compare different doxycycline treatment regimens in a robust in vivo model of inhalational Q fever for the first time. The results from these studies may be used to optimize future treatment regimens in a range of clinical scenarios and thus improve medical outcomes. 

Further developing the A/J mouse model by determining viable counts has provided new insight into Q fever. Severe headaches commonly occur in up to 87% of acute Q fever cases [26], often attributed to febrile illness and dehydration. The isolation of viable bacteria from brain tissue, however, suggests a more direct effect, not reported before. The colonization of the bone marrow has been reported before in both mice and man [22,27] and in adipose tissue in mice [24]; therefore, this study confirms these tissues’ ability to harbor bacteria with potentially long-term consequences. The blood samples were all positive by PCR from day 5 to day 8 PC (peak of disease), indicating the utility of this as an early diagnostic test for acute Q fever prior to seroconversion.

Another confirmatory finding was the presence of viable *C. burnetii* within the semen samples and the possibility of sexual transmission [20,28]. However, our images suggest that the coxiella resides within the cytoplasm of associated white cells, rather than extra-cellularly attached on the head of the sperm [28]. Small numbers of white cells are normally present in semen samples [29], and due to the method of collection, we cannot enumerate those we detected. We observed the same abnormal appearance of the sperm previously reported [28], but we suggest that it is an inflammatory response to infection. The presence of the inflammatory extracellular material on the head of the sperm would presumably cause temporary infertility [30].

The successful treatment of Q fever should be judged on two criteria: relief of acute symptoms (fever, headache, etc.) and prevention of long-term sequelae of either Q fever fatigue syndrome (QFS) or chronic Q fever (CQF). QFS is characterized by persistent fatigue following an acute Q fever infection, leading to substantial morbidity, and occurring in up to 60% of patients for periods exceeding 6–12 months [31]. Whereas CQF, occurring in up to 5% of Q fever infections, is most often presented as endocarditis, a life-threatening condition [32]. The severity of the acute Q fever (especially in cases requiring hospitalization) and the duration of symptoms are associated with a worse long-term health status including QFS [25]. However, only 38% of CQF patients recall the acute infection [32]. Both conditions are thought to be caused by bacteria persisting in immunologically privileged/sanctuary sites, although QFS appears to be driven by antigens without the presence of viable bacteria [23,33]. Our model does not cover the long-term outcomes of infection, but there is increasing evidence to suggest a connection between the failure to raise sterilizing immunity and either QFS or CFS [25].

In our model, all doxycycline treatment regimens were effective at limiting the weight loss associated with Q fever, and at reducing the occurrence of viable bacteria (to below our detection limits) in tissues such as bone marrow and adipose. Doxycycline monohydrate was better tolerated than our previously used doxycycline hyclate formulation [17], resulting in reduced drug-induced clinical signs such as fur ruffling. This may have resulted in a more successful treatment, as previously we have seen some weight loss starting around day 14 after a course of 7 days of treatment, suggestive of the resurgence of the infection. There was still some drug-related weight loss in the pre-treatment group; however, doxycycline as chemoprophylaxis in this model was also detrimental in terms of the immune response and clearance. Equally, there appeared to be no benefit from treatment after the acute infection.

The efficacy of treatment, as primarily quantified by weight loss prevention, once signs of clinical infection were noted (Figure 4) was particularly impressive, resulting in the relief of signs within 24 h of the start of treatment. In human Q fever, doxycycline has been shown to resolve fever within 2–3 days compared to a mean of 12.5 days of fever in untreated patients [34]. The optimum time to start treatment following exposure to *C. burnetii* is not known. A small study from the 1950s in humans following deliberate exposure found that oxytetracycline initiated at the point of fever onset stopped symptoms after 24–48 h [35]. Short-term (7 days) treatment started late in the incubation period but before the onset of fever prevented any symptoms from developing, whilst treatment initiated earlier, within 24 h of exposure, merely prolonged the incubation period [35], in a manner similar to the effect previously observed with 7-day doxycycline hyclate treatment for this mouse model [17]. 

Doxycycline is a bacteriostatic agent and therefore only acts to inhibit the growth of *C. burnetii*. In order to clear this intracellular bacterium, an innate and adaptive immune response is also required [35]. Therefore, treating the disease too early in the incubation period may prevent sufficient stimulation of the immune system to develop an adaptive response, and therefore lead to persistent or quiescent infection. The results of these series of studies in the A/J mice model support this theory. Treatment starting after two days of unchecked bacterial growth (i.e., before the onset of signs) was sufficient to trigger a whole immune response due to the high infectious dose required in mouse models. With-holding treatment until the occurrence of signs (day 5 rather than day 2) appears to be detrimental in this model, resulting in less bacterial clearance presumably due to the greater bacterial burden at the initiation of treatment. However, treatment with doxycycline at any time point led to a reduction in the rate of bacterial clearance compared to the control group, which may be a route for developing chronic infection.

Doxycycline also has immunosuppressive effects: recently, it has been demonstrated to limit IFNγ production by PBMCs and specifically T cells [36,37]. Good production of IFNγ is considered to be the critical first step in initiating control over *C. burnetii* growth and triggering the development of an adaptive immune response [38]. Studies in BALB/C mice suggest bacteria are cleared from day 10 onwards, coinciding with antigen-induced IFNγ production by splenocytes and the emergence of IgM phase II-antibodies [39]. In our study, circulating IFNγ in water-treated control mice peaked earlier, between day 5 and 8, so the immune activation on day 5 should have been well underway. Even starting treatment post-recovery (day 10) appeared to have had a detrimental effect on clearance. We postulate that exposure to doxycycline limited bacterial replication to a level of complete dormancy. This reduced the stimulation to the immune system, resulting in near normal levels of circulating IFNγ. In turn, this slowed the bacterial clearance, an activity provided by the innate immune response. However, once the adaptive immune response was initiated, it continued to develop, resulting in all groups appearing to have comparable *C. burnetii*-specific immune functions (CMI and antibody).

There are limitations to using a mouse model to inform the optimum time point to start treatment because the model described here is not wholly representative of human disease [10]. The majority of human infections are thought to be caused by the inhalation of just a few bacteria [8,40], it can take weeks to develop fever and around 40% of cases are symptom-free [41]; this scenario is difficult to model experimentally. *C. burnetii* is known to have a number of immune-evasion strategies [42] which might be rendered ineffectual by the high initial infection rates (as described here), leading to a relatively intense stimulation of the immune system.

Therefore, it could be hypothesized that the reduced rate of clearance might impact the development of long-term conditions. Along with slower clearance, we have some evidence of bacteria being trafficked systemically during treatment both in this study with doxycycline and previously with Finafloxacin, another bacteriostatic antibiotic [43]. Despite extending the infection model out to more than 35 days, complete clearance of the infection was not seen in any of the treated groups. It is a real possibility that delayed clearance post-treatment allowed low level systemic spread and recolonization of immune privileged sites at levels too low for our detection. Our results might suggest that in acute disease, for cases resulting in only mild clinical signs, there might be some benefit in not taking doxycycline.

This model does not explore the characteristics of chronic Q fever infection, such as endocarditis [44], and therefore cannot predict the development of chronic Q fever. It is also not able to inform on the human equivalent clinical threshold for treatment to prevent chronic infection. In the absence of being able to perform large human trials to investigate the optimal start points for doxycycline therapy, further studies utilizing the mouse model are required to identify a novel bactericidal antimicrobial agent. A bactericidal antimicrobial would significantly reduce the likelihood of recrudescence of infection after antimicrobials are stopped.

Despite the limitations of this model, the data presented here do not support the use of doxycycline as a prophylactic nor as a treatment to help clear any residual infection if the individual has recovered (and is symptom-free) from the acute infection. Instead, these studies indicate that post-exposure doxycycline treatments starting on either day 2 or day 5 were optimal, with reduced clinical signs, but also delayed the systemic clearance of viable bacteria. The earliest, but not prophylactic, treatment was the most effective. Effective clearance was dependent on the development of an adaptive immune response, but also driven by sufficient bacterial activity to maintain an active immune response. Despite this, and the fact that the animals receiving only 7 days of treatment showed no signs of relapse post-treatment, there are sufficient human data to recommend continuing with the 14-day treatment regime [35,45]. In addition, we recommend the preferential use of doxycycline monohydrate due to improved tolerance.

## 4. Materials and Methods

Mice. Groups of age-matched male 8–10-week-old (20–22 g) A/Jola mice (Envigo, Huntington, UK) were housed on a 12 h day-night light cycle, with food and water available ad libitum in an Advisory Committee on Dangerous Pathogens (ACDP, UK) level 3 compliant rigid walled isolator and allowed to acclimatize for 7 days before challenge. All procedures were conducted under a project license approved by an internal ethical review, and in accordance with both the UK Animal (Scientific Procedures) Act (1986) and the 1989 Codes of Practice for the Housing and Care of Animals used in Scientific Procedures.

Challenge. Aerosols were generated using a 3-jet Collison nebulizer, containing a volume of 20 mL of *Coxiella burnetii* NMI at a concentration of 1 × 10^8^ cfu/mL, controlled and conditioned to 50% (±5%) relative humidity, by an AeroMP platform system (Biaera Technologies, Hagerstown, MD, USA) [46]. Animals were exposed to the aerosol for a total of 10 min, with sampling achieved for 1 min at the mid-point of the challenge (4.5–5.5 min) using an all-glass impinger (AGI-30; Ace Glass, Vineland, NJ, USA) containing 10 mL phosphate-buffered saline.
$$concentration\;per\;L\;of\;air=\frac{\left(conncentration\;per\;mL\right)\times \left(sampler\;volume\right)}{\left(sampler\;flow\;rate\right)\times \left(sampling\;duration\right)}$$

A maximum of 20 animals were exposed in any single aerosol exposure, and were physically restrained in holding tubes, enabling nose-only exposure to the aerosol. The calculated, presented challenge dose was determined using the serial diluted bacterial enumerations from the aerosol samples, and Guyton’s formula for the respiratory volumes of laboratory animals (20 mL min^−1^) [47].
Presented dose=concentration per L of air×minute respiratory volume×challenge duration

A retained dose of 40% of the presented dose was also calculated [48].
Retained dose=presented dose×0.4

Treatment studies. A schematic of the studies is shown in Figure 8.

Study 1: The post-challenge groups of 15 mice were treated from 5 days post-challenge (PC), for either 7 or 14 days with oral doxycycline monohydrate dissolved in distilled water (105 mg/kg twice daily). The pre-challenge groups started treatment 24 h before challenge and continued treatment until 7 or 14 days after challenge with the same dose. Previously determined pharmacokinetic data were used to calculate that this would achieve the human equivalent dose [11]. The control groups of mice were treated with comparable volumes of distilled water (WC). At the end of the treatment period, blood and tissues were collected from 5 mice from each group. Blood was collected by cardiac puncture under inhaled anesthesia with halothane, and then mice were euthanized by cervical dislocation. The remaining mice were observed until 35 days PC. Both studies had groups of water-treated challenged controls. At cull points, organs such as the lung, spleen, brain, bone marrow, adipose (from around peritoneal cavity), testicles and blood were aseptically removed, homogenized through a 45 μm filter into 1 mL of PBS, serially diluted and cultured for viable bacteria. Bacterial DNA was also determined by PCR. One mouse was euthanized for reaching the humane end point of weight loss of greater than 25% and displaying clinical signs of a lack of responsiveness.

Study 2: Groups of 6 mice were treated 24 h pre-, 2 days, 5 days or 10 days post-challenge (PC), and this continued for 14 days (15 days for the pre-treatment group) with doxycycline monohydrate (105 mg/kg twice daily). The control groups of challenged and unchallenged mice were treated with comparable volumes of water (WC). Mice were observed for clinical signs (ruffled skin, arched back, dehydration and wasp-waisted appearance) and weighed daily for a further 14 days after the cessation of treatment days, when they were euthanized by cervical dislocation and the lungs and spleens removed for viable counts.

Sperm microscopy. Excised testicles were aseptically cut into pieces in 1 mL of PBS above a 45 μm filter and left for 20 min to allow the sperm to swim through the filter. A 50 μL subsample was taken from below the filter for microscopy, and the remaining sample was recombined and homogenized as above to provide a viable count. The microscopy subsample was inactivated in 4% PFA, permeabilized (BD Biosciences, Oxford, UK) and stained with the FITC-conjugated anti-*Coxiella* antibody (BBI, Salisbury, UK) and the nuclear stain, DAPI (Life technologies, Paisley, UK). Samples were viewed with a Zeiss LM-10 fluorescent microscope.

Bacteria. *C. burnetii* NMI (RSA493) was cultured axenically in ACCM-2 (Sunrise Science Products, San Diego, CA, USA) for 7 days, and shaken at 75 rpm in a sealed container with a GENbox microaer atmosphere generator (bioMérieux, Marcy-l’Étoile, France) to produce challenge stock [11]. The bacteria were centrifuged at 10,000× *g* for 20 min, resuspended in PBS at a concentration of approximately 1 × 10^9^ cfu/mL and stored frozen. For enumeration from tissues, samples were plated on ACCM-2 agar, supplemented with 0.5 mM tryptophan [49] and incubated statically at 37 °C for 10 days (in 5% CO_2_ and 2.5% O_2_). The limit of detection (LOD) was dependent on organ size and was approx. 200 cfu/g in study 1 and 20 cfu/g in study 2, due to increased plating to determine clearance. Counts were also determined by PCR in study 1 using the com1 gene [17]. Challenge enumeration post-aerosolization was determined by PCR.

Enumeration by PCR to calculate genome equivalents (GE). *C. burnetii* was enumerated using RT-PCR targeting the *com1* gene (forward primer, CGACCGAAGCATAAAAGTCAATG; reverse primer, ATTTCATCTTGCTCTGCTCTAACAAC; probe, TTATGCGCGCTTTCGACTACCATTTCA). The probe was covalently labelled at the 5′ end with the reporter dye FAM and at the 3′ end with the quencher dye BHQ-1. The primers and probe were purchased (ATDBio Southampton UK). Chromosomal DNA was extracted by using the Qiagen QIAmp DNA mini kit/blood and tissue (Qiagen Manchester UK). The RT-PCR comprised 12 µL template DNA, forward primer (900 nM), reverse primer (300 nM), probe (200 nM) and PCR master mix containing 0.04 U JumpStart Taq DNA polymerase 21 µL (Sigma-Aldrich, Gillingham, UK), 0.2 mM dNTPs, 8% *w*/*v* glycerol, 4 mM MgCl2, 50 mM Tris/HCl, 1 mg BSA ml21 and 0.5 mM EGTA. PCR cycling conditions comprised 3 min at 95 °C and 30 s at 60 °C, followed by 50 two-step cycles of 15 s at 95 °C and 30 s at 60 °C. Standard curves were made by spiking naïve tissue/blood with serial dilutions of viable bacteria, and extracted as above.

Serum IFNγ quantification. Serum samples were stored frozen at −80 °C until analyzed. The levels were determined using the ProcartaPlex™ multiplex immunoassay kit (ThermoFisher, Paisley, UK), used according to the manufacturer’s instructions and read on a Luminex™ machine. 

Antibody determination. Total IgG was quantified by ELISA from sera taken on day 35 PC. Coxevac (inactivated veterinary *Coxiella burnetii* vaccine, European Medicines Agency, The Netherlands) was used for the antigen to coat 96-well plates (50 μL/well diluted 1:2 in carbonate coating buffer (Sigma-Aldrich, Gillingham, UK) to give approx. 1 × 10^8^ GE/mL). The ELISA was performed in the standard manner, briefly: plates were incubated at 4 °C overnight, blocked with 2% skimmed milk powder (in PBS with 0.5% Tween (Sigma, Gillingham, UK), washed and incubated (at 37 °C for 2 h) with a dilution series of serum starting at 1:50. Plates were washed again, and antibody binding was detected by mouse anti-IgG horseradish peroxidase (Sigma-Aldrich, Gillingham, UK). The mean of duplicate values was considered positive if greater than the mean + 2 SD of the negative sera. 

Cell-mediated restimulation. Splenic tissue taken on day 35 PC was gently homogenized and passed through a 40 µm sieve into 1 mL of sterile Leibovitz’s L-15 medium supplemented 2 mM L-glutamine, 100 U/mL penicillin, 100 µg/mL streptomycin and 10% (*v*/*v*) FCS (all Sigma-Aldrich, Gillingham, UK). Splenocytes were stimulated with either concanavalin A (ConA, Sigma-Aldrich, Gillingham, UK) at a final concentration of 2.5 µg/mL, Coxevac (inactivated veterinary *Coxiella burnetii* vaccine, European Medicines Agency, The Netherlands) concentration of 1 × 10^8^ GE/mL or media as a negative control. Twenty-five microliters per well of the splenocytes was added to 175 µL of stimulants in the complete L-15 medium and incubated at 37 °C for 24 h, and the supernatant frozen at −80 °C until assayed for IFN-γ production by ELISA (MABTEC, Nacka Strand, Sweden), following the manufacturer’s instructions.

Statistical analysis. Significant differences in the levels of secreted IFNγ or antibody production were determined by ANOVA, and differences in weight loss profiles were determined by a two-way RM ANOVA with Tukey’s multiple comparisons. The statistical analysis was performed by Graph Pad Prism version 8.

## Figures and Tables

**Figure 1 antibiotics-12-00914-f001:**
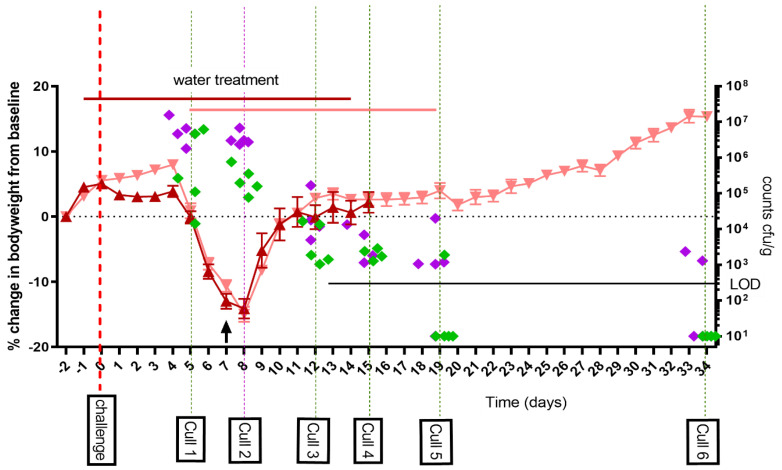
Weight loss in A/J mice following an aerosol challenge of *C. burnetii* of a mean retained dose of 6.7 × 10^4^ GE and treated with 50 μL of water twice daily orally starting either from 1 day prior (dark red triangle) or from day 5 post-challenge (pale red triangle) to 14 days PC. Data shown are the mean percentage weight loss from baseline +/− standard error of the mean (SEM). The arrow represents the time point where 1 mouse was euthanized due to reaching a humane end point. Five mice were euthanized at each of the marked time points and selected tissues homogenized and cultured to obtain viable counts; lung (green diamond) and spleen (purple diamond) expressed as cfu/g of tissue. LOD: limit of detection.

**Figure 2 antibiotics-12-00914-f002:**
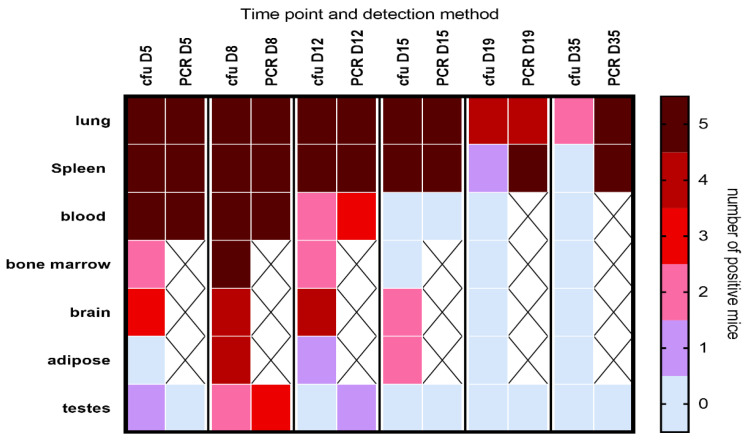
Heat map illustrating the dissemination of *C. burnetii* in A/J mice challenged via the aerosol route with a mean retained dose of 6.7 × 10^4^ GE of *C. burnetii*. The heat map shows the number of mice in which bacteria were detected by either culture (cfu) or by PCR at different time points. The groups were provided with water but no antibiotic therapy. X denotes sample not tested. LOD was approx. 20 cfu/spleen, 10 cfu/lung, 20 cfu/mL blood and below 50 cfu for all other tissues; detection by PCR was approx. 50 GE/tissue.

**Figure 3 antibiotics-12-00914-f003:**
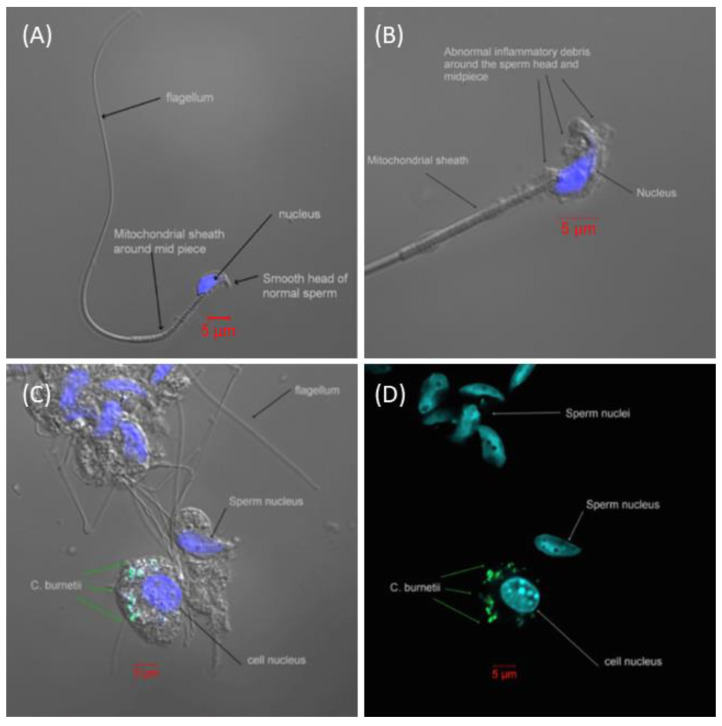
Spermatozoa from male A/J mice visualized by confocal microscopy. (**A**) Representative image of healthy spermatozoa from an unexposed A/J mouse. (**B**) Representative image of spermatozoa from a mouse exposed to an aerosol challenge of a mean retained dose of 6.7 × 10^4^ GE of *C. burnetii* on day 8 PC. There is abnormal inflammatory debris over the head and midpiece of the spermatozoa, but no evidence of *C. burnetii*. (**C**) A collection of spermatozoa and surrounding cells from within the testicles. *C. burnetii* can be seen within an adjacent cell but no *C. burnetii* attached to the spermatozoa. (**D**) As image C without light image, more clearly demonstrating the bacteria in the adjacent cell and not attached to the spermatozoa. (Blue = DAPI, DNA stain; green = mouse anti-*C. burnetii* LPS monoclonal FITC stain; space bar in red = 5 µm).

**Figure 4 antibiotics-12-00914-f004:**
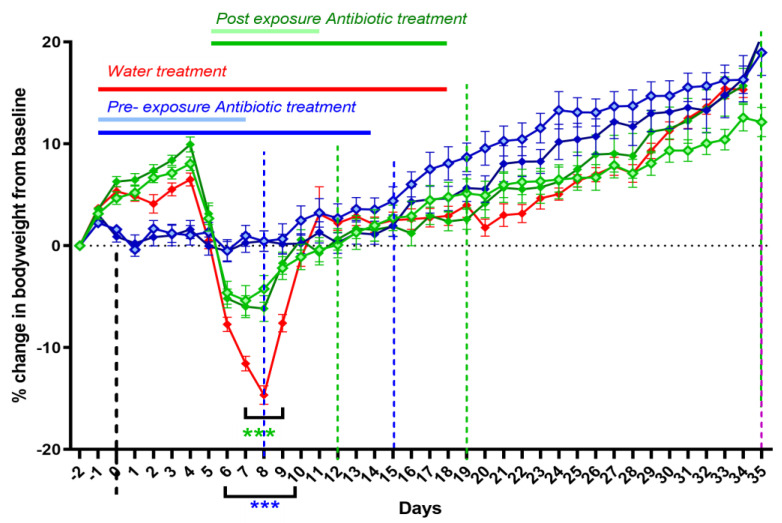
Weight loss in A/J mice challenged via the aerosol route with a mean retained dose of 6.7 × 10^4^ GE of *C. burnetii*, and 7 or 14 days treatment with doxycycline monohydrate (DM) starting on day −1 (pale blue diamonds: 7 days, solid blue diamonds: 14 days treatment) or day 5 (pale green diamonds: 7 days, solid green diamonds: 14 days treatment) PC. Red line: water-treated control group (WC). Groups of 5 were euthanized at 8, 12, 15 and 19 days PC (dotted lines) and 10 at end of study time points. Statistically significant differences in the weights compared to the control group were determined by a two-way RM ANOVA with Tukey’s multiple comparisons where *** *p* < 0.001.

**Figure 5 antibiotics-12-00914-f005:**
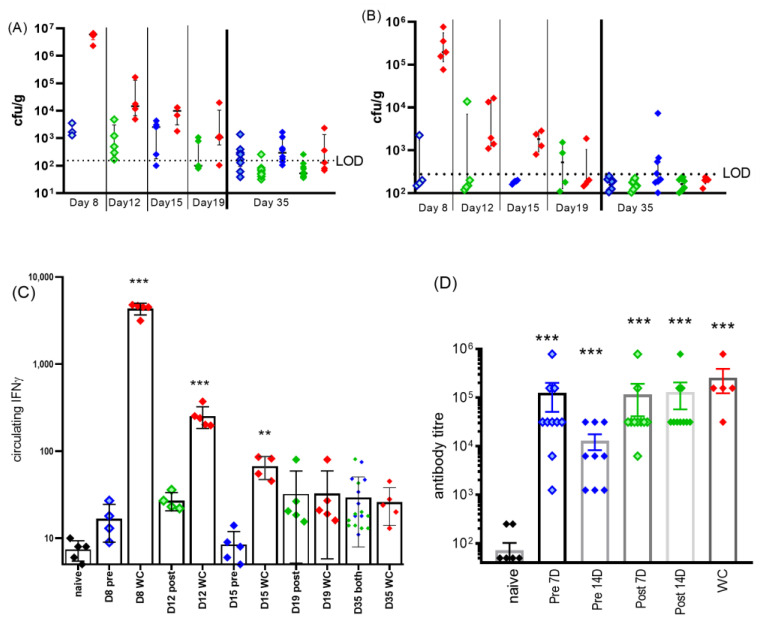
Clinical detail of *C. burnetii*-challenged, doxycycline-treated mice. Bacterial colonization in (**A**) lung and (**B**) spleen after treatments and at study end (day 35). LOD was approx. 20 cfu/organ. Pre-treated blue diamonds (pale blue: 7 days, solid blue: 14 days treatment) or treatment starting on day 5: green diamonds (pale green: 7 days, solid green: 14 days treatment). WC (water-treated control) group: red diamonds. Groups were 5 for inter-study time points and 10 for end of study. (**C**) Corresponding levels of circulating IFNγ, significance from naïve samples by ANOVA, ** *p* < 0.01 and *** *p* < 0.001. (**D**) IgG antibody titers to phase I *coxiella* (coxevac) on day 35. All challenged groups produced significant levels of antibody above the background present in naïve sera (*p* < 0.001).

**Figure 6 antibiotics-12-00914-f006:**
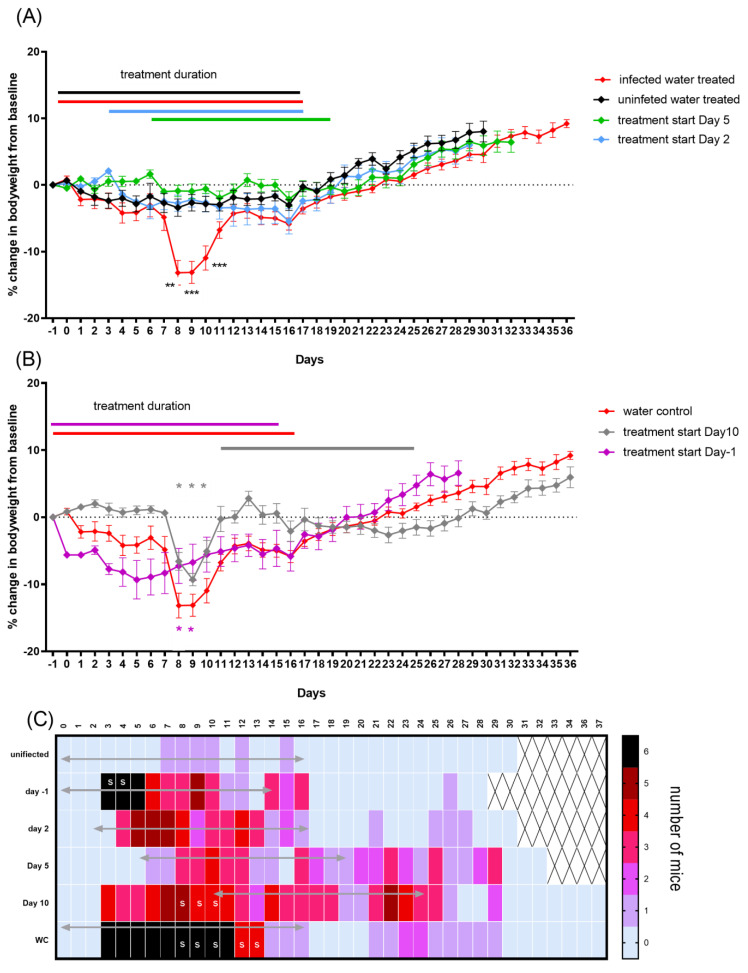
*Coxiella* infection-associated signs in A/J mice following an aerosol challenge with a mean retained dose of 2.9 × 10^4^ GE of *C. burnetii* and treatment initiated at −1, 2, 5 or 10 days post-challenge. (**A**) Weight loss as a percentage of baseline for water-treated challenged control group (red line) and unchallenged water-treated control group (black line). Doxycycline treatment starting on day 2 (blue) and day 5 (green). (**B**) Weight loss for treatment starting on day 10 (grey line) and one day before challenge (purple) compared to challenged water-treated group. (**C**) Number of mice per group showing mild clinical signs of ruffled fur. S denotes mice presenting with more severe signs such as hunched posture. The mean weight changes from baseline with standard error of the mean (SEM) are presented. Significant differences in weights were determined by a two-way RM ANOVA with Tukey’s multiple comparisons; *p* < 0.01 ** and *p* < 0.001 ***.

**Figure 7 antibiotics-12-00914-f007:**
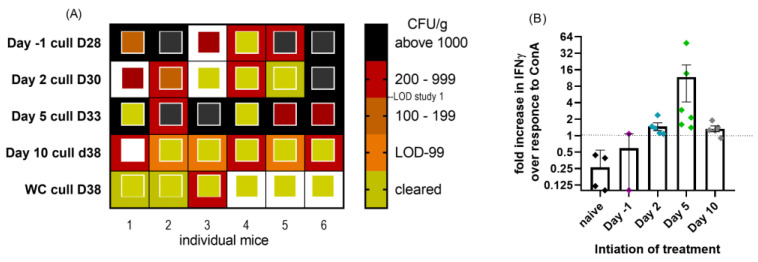
(**A**) Heat map to show level of bacterial colonization of lung and spleen samples from mice exposed to *C. burnetii* and treated with doxycycline initiated at −1, 2, 5 or 10 days post-challenge for 14 days. Samples were taken 14 days after treatment ended. The outside box is lung cfu/g, and the inside smaller box represents spleen cfu/g. LOD was 20 cfu/g (2–5 cfu/organ). White box denotes data not available. (**B**) Cell-mediated immunity determined from the spleens of the same mice. Point colours to match Figure 6, treatment start day −1 purple, day 2 blue, day 5 green and day 10 grey PC. Fold increase in IFNγ production by spleen homogenates following exposure to heat-killed phase I *coxiella* compared to ConA control stimulant.

**Figure 8 antibiotics-12-00914-f008:**
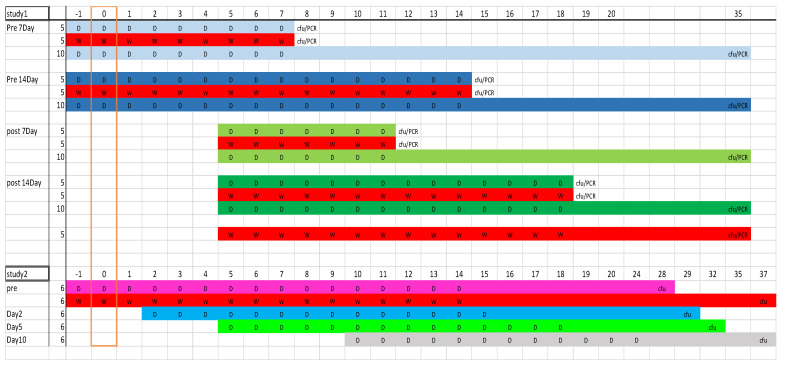
Schematic of the mouse studies 1 and 2 illustrating dosing schedules and sample points (D = doxycycline and W = water control); group colors match those used in results. Numbers along the top denote time point (in days) with the challenge occurring on day 0. Column 2 gives number of mice per group.

## Data Availability

None available.

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
