# Peer review of "Evaluation of Alternative Doxycycline Antibiotic Regimes in an Inhalational Murine Model of Q Fever"

_antibiotics, 2023, doi:10.3390/antibiotics12050914_

Round 1

Reviewer 1 Report

Authors evaluated the efficacy of doxycycline monohydrate regimens in an inhalational murine model of Q fever caused by Coxiella burnetii. This study describes the effectiveness of doxycycline against Q fever along with its immunomodulatory potential by significantly altering the level of g IFNg and antibodies titer. The study also suggests the clearance of viable bacteria from tissue & significantly impact on immune organs. The study is interesting and the manuscript is comprehensive. However, I suggest following minor changes to improve the quality and significance of the study.

Comments:

1.      In the Abstract section, authors should indicate the key tissues where doxycycline delayed the clearance of viable bacteria. Abstract is the lack of key findings of the study. Authors should add significant results in the abstract section as well.

2.      In the Methodology section, what were the criteria of administration of water in the control group? Why did the authors not select normal saline or other solvents? Secondly, the water used in this study was distilled water or tap. Water?

3.      It would be more effective if authors provide justification for the use of various types of treatment time and further explain how each parameter measured correlates to outcome of the study.

4.      The safety profile of the drug should be discussed.

5.      Please add software specification used in statistical analysis (ANOVA, Graph prism)

6.      As the bacterium (Coxiella burnetii) is intracellular, the authors should address phagocytic index which is the main property of macrophages, key player of cell mediated immunity.

7.      Limitation should be discussed in the limitation section.

8.      Authors should describe the Future perspective and clinical significance of the study.

9.      Authors should add abbreviation list in the manuscript.

Author Response

  1. In the Abstract section, authors should indicate the key tissueswhere doxycycline delayed the clearance of viable bacteria. Abstract is the lack of key findings of the study. Authors should add significant results in the abstract section as well.

Yes, all of the reviewers made these kind of comments. We have re-written the abstract to include more of the findings of the study. We have changed key tissues to spleen tissue which is the tissue we were using as an indicator of systemic infection.

  1. In the Methodology section, what were the criteria of administration of water in the control group? Why did the authors not select normal saline or other solvents? Secondly, the water used in this study was distilled water or tap. Water?

The control group were given water as this was used to dilute the doxycycline. It was sterile distilled water (which is also what the mice are given to drink normally). This has all been clarified in the methods section.

  1. It would be more effective if authors provide justification for the use of various types of treatment time and further explain how each parameter measured correlates to outcome of the study.

Agreed. We have expanded on the treatment start time and treatment lengths in the introduction (line 93 onwards)

  1. The safety profile of the drug should be discussed.

Doxycycline has been used to treat Q fever since the 1960s, so really needs no introduction. In the UK the formulation most commonly used is hyclate (for financial reasons), but is frequently reported as causing gastro-intestinal side effects. It is very poorly tolerated by the mice, so in the interests of animal welfare we moved to the monocyte formulation. We have added some text to the introduction to make our reasoning for the switching of doxycycline more obvious

  1. Please add software specification used in statistical analysis (ANOVA, Graph prism)

Added to the methods section

  1. As the bacterium (Coxiella burnetii) isintracellular, the authors should address phagocytic index which is the main property of macrophages, key player of cell mediated immunity.

The reviewer is right, this would be an interesting feature to have assessed, however it was beyond the scope of this study

  1. Limitation should be discussed in the limitation section.

We have expanded on the limitations of this study, but rather than having a spate section we have included this as part of the discussion

  1. Authors should describe the Future perspective and clinical significance of the study.

We have expanded on the clinical significance in the discussion section, but are wary of over interpreting mouse data for human treatment

  1. Authors should add abbreviation list in the manuscript.

We have tried to make our abbreviations clearer rather than have a list. We have included a schematic of the studies (in the methods section) to help readers interpret our groups

Reviewer 2 Report

The submitted manuscript aims to evaluate efficacy of various antibiotic treatment regimens in a murine model of inhalational Q fever. Further, the authors described bacterial dissemination and clearance for various treatment groups. The authors concluded that early antibiotic treatment was effective in controlling disease.

The manuscript addresses a timely topic which fits within the scope of Antibiotics. The data is valuable and of interest to the field; however, I feel that the manuscript requires major changes to aid in reader comprehension and data integrity. I found the data presentation style confusing and cumbersome. Further, the text has many formatting errors and missing data/references. I feel that improved presentation of the data, rectifying formatting errors, and better explaining the data would greatly enhance the manuscript. Indeed, I feel that these changes are obligatory to improve the reader’s comprehension and the validity of the data presented. Major and minor comments are as follows:

Major Comments:

Abstract: The abstract was quite sparse and lacked context in the form of introductory information. Further, I feel that important conclusions were omitted.

The manuscript contained many formatting errors that impede reader comprehension. For example: spacing issues, improper punctuation, lack of periods, and random parentheses.

I was unable to find in-text citations for several figures. For example, figure 3A, C, and D. Further, figure citations (e.g., A, B, C) were not in order in the text. Lastly, some data that was discussed seemed to lack a corresponding figure (e.g., splenomegaly data from lines 106-108 and 177-187). I was also confused by line 175 and the citation of figure 5A, which does not seem to be represented in this figure.

I had trouble discerning many of the figures and suggest reformatting them. Specifically, Figure 1 appears cumbersome and I had trouble parsing out treatment groups. I suggest creating two separate graphs for body weight and organ CFU. Figures 5A and 7A may also be better represented by separate figures per organ. The statistical analysis of figure 5B appears incomplete, as distinct treatment groups were not compared to each other.

As a non-quantitative endpoint, I feel that the inclusion of ruffled fur in graphical form is a bit odd. Perhaps this could be omitted or moved into the supplemental information, as the weigh loss data is more convincing to me.

I feel that the presentation of spermatozoa images was a bit random. Perhaps some context regarding this tissue could be included in the introduction, as no other organs were visualized. Further, I suggest describing magnification, adding scale bars, and describing if these are representative images per group.

Minor Comments:

Line 27: I suggest defining GE as "genome equivalents" at the first use.

Lines 58-59: The study described should be cited.

Line 64: I would argue that guinea pig fatality in the infection model can occur, but at comparably high doses. Obviously, this depends on route and animal strain. Murine fatality can also occur. I suggest clarifying this statement.

I suggest changing “culled” to “euthanize”.

Line 118: “Humane end point” criteria should be defined in the materials and methods.

Figure 4: The treatment bar legends are overlapping and should be presented more clearly.

Figure 6A: The untreated:water group does not gain weight throughout the treatment duration. I suggest explaining this further.

Line 466: Were IgM and/or phase II anti-C. burnetii levels assayed? Given the various euthanasia timepoints, this information might be informative.

Funding, acknowledgements, and conflict of interest sections: These sections have not been completed.

Author Response

Abstract: The abstract was quite sparse and lacked context in the form of introductory information. Further, I feel that important conclusions were omitted.

Yes, all of the reviewers made these kind of comments. We have re-written the abstract to include more of the findings of the study.

 The manuscript contained many formatting errors that impede reader comprehension. For example: spacing issues, improper punctuation, lack of periods, and random parentheses.

Yes, the reformatting caused some errors, we do apologise, hopefully it reads better now

I was unable to find in-text citations for several figures. For example, figure 3A, C, and D.

Yes, sorry, these have now been added in

Further, figure citations (e.g., A, B, C) were not in order in the text. Lastly, some data that was discussed seemed to lack a corresponding figure (e.g., splenomegaly data from lines 106-108 and 177-187). I was also confused by line 175 and the citation of figure 5A, which does not seem to be represented in this figure.

Yes, these were badly worded, and have been changed to make it clearer

I had trouble discerning many of the figures and suggest reformatting them. Specifically, Figure 1 appears cumbersome and I had trouble parsing out treatment groups. I suggest creating two separate graphs for body weight and organ CFU.

We accept that figure 1 has a lot of information included, however we would like to keep it unchanged, as it allows the reader to make a direct assessment of how organ bacterial counts are related to weight loss

 Figures 5A and 7A may also be better represented by separate figures per organ.

Yes all of the reviewers found 5A difficult to follow. It has now been split into 2 separate graphs. Figure 7A we would like to keep as it is, as it allows the counts in the lung to be directly compared to the counts in the spleen for each individual animal

The statistical analysis of figure 5B appears incomplete, as distinct treatment groups were not compared to each other.

Yes, again this transferred incorrectly. The statistical marking has now been redone

As a non-quantitative endpoint, I feel that the inclusion of ruffled fur in graphical form is a bit odd. Perhaps this could be omitted or moved into the supplemental information, as the weigh loss data is more convincing to me.

We were keen to try and determine if treatment starting day 2 was better than day 5. The weight loss data suggested day 5 was marginally better, whereas the clinical signs and bacterial clearance suggested otherwise. We would like to leave it in to allow the reader to draw their own conclusions

I feel that the presentation of spermatozoa images was a bit random. Perhaps some context regarding this tissue could be included in the introduction, as no other organs were visualized.

Yes, we have expanded on our interests in this tissue within the introduction

Further, I suggest describing magnification, adding scale bars, and describing if these are representative images per group.

Yes, we have added  

Minor Comments:

Line 27: I suggest defining GE as "genome equivalents" at the first use.

Good suggestion. Done

Lines 58-59: The study described should be cited.

Yes, that was an omission, now added

Line 64: I would argue that guinea pig fatality in the infection model can occur, but at comparably high doses. Obviously, this depends on route and animal strain. Murine fatality can also occur. I suggest clarifying this statement.

We have used the words frequently and often, but feel it is fair to describe the typically used guinea pig model as too severe and the mouse as too mild.

I suggest changing “culled” to “euthanize”.

Yes that’s better, we have changed

Line 118: “Humane end point” criteria should be defined in the materials and methods.

Good point, we have now added this

Figure 4: The treatment bar legends are overlapping and should be presented more clearly.

Again good point, we have changed this

Figure 6A: The untreated:water group does not gain weight throughout the treatment duration. I suggest explaining this further.

We frequently see some minor detrimental effect caused by constant handling, we have added this as a limitation in the discussion.

Line 466: Were IgM and/or phase II anti-C. burnetii levels assayed? Given the various euthanasia timepoints, this information might be informative.

Yes, with hindsight we can see that this would have added information, however it wasn’t done

Funding, acknowledgements, and conflict of interest sections: These sections have not been completed.

Now filled out

Reviewer 3 Report

Abstract. This is too short and does not do justice to the manuscript. The authors must provide more details in there for the benefit of people interested to read it.

Introduction. In a final paragraph, please summarise the hypothesis upon which the work has been based. Also, please present clearly the objectives of the study.

M&M. Please include a table with details of the design of the two experiments. This will make understanding of the design easier.

All the details of the PCRs must be included in a table.

Discussion. This is more shallow than I have expected after such a work. Please revise and improve to cover all facets of the study and please go deeper into mechanisms and interactions.

Overall. Good work, but the presentation is not yet to publication standards. Extensive improvement and re-evaluation.

Author Response

Abstract. This is too short and does not do justice to the manuscript. The authors must provide more details in there for the benefit of people interested to read it.

Yes, all of the reviewers made these kind of comments. We have re-written the abstract to include more of the findings of the study.

Introduction. In a final paragraph, please summarise the hypothesis upon which the work has been based. Also, please present clearly the objectives of the study.

The treatment start times and durations were designed to mimic human scenarios. We have expanded our reasoning on the treatment start time and treatment lengths in the introduction (line 93 onwards).

M&M. Please include a table with details of the design of the two experiments. This will make understanding of the design easier.

Yes, good idea. We have now included a schematic showing the dosing durations, start points and culls for each of the groups to help clarify the 2 studies

All the details of the PCRs must be included in a table.

PCR enumeration was only preformed in the first study, the details are in figure 2

Discussion. This is more shallow than I have expected after such a work. Please revise and improve to cover all facets of the study and please go deeper into mechanisms and interactions.

We have attempted to improve the discussion, but are wary of over interpreting mouse data for human treatment. We have more fully discussed the limitations of these studies

Reviewer 4 Report

Overall this is a well-written manuscript describing an aerosol model for Coxiella in A/J mice and the effect or pre and post-treatment with antibiotics.

Was a MIC done for in vitro grown Coxiella? how was the dosage of 105 mg/kg twice daily determined? It is surprising that the antibiotic of choice used in the treatment of Coxiella infections in humans had a deleterious effect in mice. If a more common form of doxycycline is used are bacterial numbers reduced? Is a prophylaxis dose less than a normal dose? This was not clearly articulated in the manuscript.

Line 208 mention a previous study that showed a reduction of bacterial load however there is no reference so it is hard to assess whether this statement is true or not.

Figure 5A is confusing it may be worth separating the data a little more. For example, the WC would have its own lung and spleen columns as a frame of reference and then maybe it would be easier to understand the other data categories

Line 204 indicates that 16-day treatment groups have significantly lower antibodies (compared to what is not mentioned) and this is not indicated in the figure itself. 

Figure 7A again is confusing is there any other way to present this data? In 7B why was a WC not included in this data? One may assume that if the Coxiella were cleared it should have the highest IFN response.

Line 379 there is a ( but not a ) 

Author Response

was a MIC done for in vitro grown Coxiella?

Yes we have published this previously, but some details of this have been added

how was the dosage of 105 mg/kg twice daily determined?

Again, previously published, references have been added

 It is surprising that the antibiotic of choice used in the treatment of Coxiella infections in humans had a deleterious effect in mice.

Doxycycline hyclate is known to cause gastrointestinal upset, and instruction for taking it often include a recommendation to stay upright for an hour afterwards, so perhaps this is due to the relatively prostrate posture of the animal? In the interests of animal welfare we moved to the monocyte formulation. We have added some text to the introduction to make our reasoning for the switching of doxycycline more obvious

If a more common form of doxycycline is used are bacterial numbers reduced?

No we have tested this, details have now been added

Is a prophylaxis dose less than a normal dose? This was not clearly articulated in the manuscript.

Sorry, the dose used was the same, we have clarified this in the methods

Line 208 mention a previous study that showed a reduction of bacterial load however there is no reference so it is hard to assess whether this statement is true or not.

Sorry that was badly worded. We were referring to the first study in this paper. We have changed the wording to make it clearer

Figure 5A is confusing it may be worth separating the data a little more. For example, the WC would have its own lung and spleen columns as a frame of reference and then maybe it would be easier to understand the other data categories

Yes all of the reviewers found 5A difficult to follow. It has now been split into 2 separate graphs

Line 204 indicates that 16-day treatment groups have significantly lower antibodies (compared to what is not mentioned) and this is not indicated in the figure itself. 

Sorry this copied over wrongly, hopefully it is clearer now, and the figure legend has been changed to help

Figure 7A again is confusing is there any other way to present this data?

We accept that Figure 7A is a little difficult to follow, but we would like to keep as it is, as it allows the counts in the lung to be directly compared to the counts in the spleen for each individual animal

In 7B why was a WC not included in this data? One may assume that if the Coxiella were cleared it should have the highest IFN response.

Yes it was unfortunate but this group was omitted from the assay. However as the assay was performed to determine how the different doxycycline treatments were affecting the animals ability to form cell mediated responses, we feel it provides useful information 

Line 379 there is a ( but not a ) 

Sorry I can’t work this out?

Round 2

Reviewer 2 Report

I feel that the manuscript has been greatly improved following revisions.

Author Response

I feel that the manuscript has been greatly improved following revisions.

Thank you 

Reviewer 3 Report

Before final acceptance, can the authors, please, discuss the clinical significance of their findings?

A brief paragraph at the end of the discussion will do.

Thank you.

Author Response

Before final acceptance, can the authors, please, discuss the clinical significance of their findings?

A brief paragraph at the end of the discussion will do.

We believe that we have discussed the clinical significance of our findings throughout the discussion and we wish to avoid any unnecessary repetition. Examples of where we have already provided discussion on this area are below.

In addition, we do accept that we have omitted any comment on the duration of treatment or formulation of doxycycline, so we have added a few lines at the end (459-462):

“Despite this and the fact that the animals receiving only 7 days of treatment showed no signs of relapse post treatment, there is sufficient human data to recommend continuing with the 14 day treatment regime (35,45). In addition, we recommend preferential use of doxycycline monohydrate due to improved tolerance

Examples of our existing discussion of clinical significance:

For example lines 382-384 we state:

doxycycline as chemoprophylaxis in this model was also detrimental in terms of the immune response and clearance. Equally there appeared to be no benefit from treatment after the acute infection”.

Again on lines 451-456 we re-iterate this by saying:

 “the data presented here does not support the use of doxycycline as a prophalytic nor as a treatment to help clear any residual infection if the individual has recovered (and is symptom free) from the acute infection. Instead, these studies indicate that post-exposure doxycycline treatments starting on either day 2 or day 5 were optimal, with reduced clinical signs, but also delayed the systemic clearance of viable bacteria. Earliest, but not prophylactic treatment was most effective”.

We have also discussed the limitations of the model and the difficulty of using it to instruct on human disease (lines 425-432) by stating:

There are limitations to using a mouse model to inform the optimum time point to start treatment because the model described here is not wholly representative of human disease [10]. The majority of human infections are thought to be caused by inhalation of just a few bacteria [8,40], can take weeks to develop fever and around 40% of cases are symptom free [41]; this scenario is difficult to model experimentally. C. burnetii is known to have a number of immune evasion strategies [42] which might be rendered ineffectual by the high initial infection rates (as described here) leading to a relatively intense stimulation of the immune system”.

We state that before drawing our most significant conclusion (lines 440-442):

Our results might suggest that in acute disease, for cases resulting in only mild clinical signs, there might be some benefit in not taking doxycycline”.

We also state that the mouse model would have most utility in (line 448-450):

“Identifying a novel bactericidal antimicrobial agent, which would significantly reduce the likelihood of recrudescence of infection after antimicrobials are stopped”.